# Explaining Soft-Goal Conflicts through Constraint Relaxations

**Rebecca Eifler**[1] , **Jeremy Frank**[2] and **Jörg Hoffmann**[1,3]

[1]Saarland University, Saarland Informatics Campus, Germany
[2]NASA Ames Research Center, Mountain View, CA, USA
[3]German Research Center for Artificial Intelligence (DFKI), Saarbrücken, Germany
{eifler, hoffmann}@cs.uni-saarland.de, Jeremy.D.Frank@nasa.gov,

## Abstract

Recent work suggests to explain trade-offs between soft-goals in terms of their *conflicts*, i. e., minimal unsolvable soft-goal subsets. But this does not explain the conflicts themselves: Why can a given set of soft-goals not be jointly achieved? Here we approach that question in terms of the underlying constraints on plans in the task at hand, namely resource availability and time windows. In this context, a natural form of explanation for a soft-goal conflict is a minimal constraint relaxation under which the conflict disappears ("if the deadline was 1 hour later, it would work"). We explore algorithms for computing such explanations. A baseline is to simply loop over all relaxed tasks and compute the conflicts for each separately. We improve over this by two algorithms that leverage information – conflicts, reachable states – across relaxed tasks. We show that these algorithms can exponentially outperform the baseline in theory, and we run experiments confirming that advantage in practice.

## 1 Introduction

Imagine planning the next Mars Rover Mission. Due to the rovers' limited resources and timing constraints for data collection and uploads, only some of the rovers' tasks can be planned. Recent work by Eifler et al. [2020a; 2020b] suggests explaining trade-offs between such soft goals in terms of *conflicts*, i. e., minimal unsolvable soft-goal subsets. However, this does not give further insights into *why* the soft-goals can not be jointly achieved. Understanding the cause of these conflicts and possible resolutions is crucial to reasoning about different options and finding the best trade-offs.

Here, we explore the question of what causes soft-goal conflicts in tasks with constraints such as resource availability and time windows. In this context, soft-goal conflicts can naturally be explained by identifying the minimal constraint relaxations under which the conflict disappears. For example the conflict {x-ray image, soil sample} could be explained by: "The rover needs 2 more units of energy or the upload window to relay X needs to be 3 time units longer, to perform both tasks". A similar approach is used in constraint programming [Lauffer and Topcu, 2019;

Senthooran *et al.*, 2021], where they introduce soft constraints, to provide suggestions on how to modify an unfeasible subset of constraints to make it feasible. We investigate algorithms for computing such explanations based on minimal relaxations in a set of given relaxations, which we instantiate with resource and time window constraint relaxations.

Eifler et al. [2020b] introduced an algorithm that computes all minimal unsolvable goal subsets of a task by expanding the whole search space while tracking all maximal solvable goal subsets. To reduce the search space size they prune all states from which, according to a given heuristic, no superset of the incumbent solution is reachable. The basic adaption of this procedure is to iteratively call it for each relaxed task separately and compute the minimal relaxed task where each conflict disappears in a post-processing step.

We introduce two algorithms that improve over this baseline by exploiting the fact that information like reachable goal subsets and states can be propagated from one relaxed task to another if the latter is more relaxed. The first algorithm, *Internal Constraint Reuse* (ICR), iteratively computes the conflicts for each increasingly relaxed planning task and reuses the reachable subgoals from less relaxed tasks. This provides the pruning function with a growing set of reachable subgoals that it can use to prune parts of the search space that do not contain any subgoals that have not yet been achieved. The second algorithm, *Search Space Reuse* (SSR), reduces duplicate work by iteratively increasing one search space instead of generating a new one per relaxed task. This is done by storing the search frontier for each task, and using it as the starting point for more relaxed tasks. Thus for each relaxed task, only the newly reachable states are generated.

We show that these algorithms can exponentially outperform the baseline, with respect to the number of generated states, in theory. Experiments on 4 resource-centric domains and 3 domains with time windows show that both algorithms perform significantly better than the baseline in practice, and that they are complementary to each other in terms of finding explanations on resource- and time-centric domains.

## 2 Preliminaries

### 2.1 Planning Formalism

A *finite-domain representation* (FDR) [Bäckström and Nebel, 1995] planning task with *soft-goals* is a tuple $\tau \ =$

$(V, A, I, G^{\text{hard}}, G^{\text{soft}})$, where $V$ is a finite set of state **variables** $v$ with domain $\mathcal{D}(v)$, $A$ is a finite set of **actions**, and $I$ is a complete assignment to $V$ called **initial state**. $G^{\text{hard}}$ and $G^{\text{soft}}$ are disjoint partial assignment to $V$ called **hard** and **soft-goal**. A **state** is a complete assignment to $V$. Variable-value pairs $v = d$ are referred to as **facts**, and (partial) variable assignments are identified by sets of facts. The value of $v$ in the (partial) variable assignments $s$ is referred to as $s(v)$. Each action consists of a **precondition** and **effect** $(\text{pre}_a, \text{eff}_a)$ defined as partial assignments to $V$. An action $a$ is **applicable** in state $s$ ($appl(a, s)$) if $\text{pre}_a \subseteq s$. Applying $a$ to $s$, denoted by $s[[a]] = s'$, changes the values of $s$ to $s'(v) := \text{eff}_a(v)$ if $\text{eff}_a(v)$ is defined and $s'(v) := s(v)$ otherwise. The resulting state of an iteratively applicable action sequence $\pi$ is denoted by $s[[\pi]]$. A **plan** is an action sequence $\pi$ where $G^{\text{hard}} \subseteq I[[\pi]]$. It achieves the soft-goals $G^{\text{soft}} \cap I[[\pi]]$. Instead of a cost-function with an upper bound, as in oversubscription planning by [Smith, 2004; Domshlak and Mirkis, 2015], here we consider constraints such as resources and time as a limiting factor for achievable soft-goals. The prefix $a_0 \cdots a_i$ of plan $\pi = a_0 \cdots a_i, a_j \cdots a_n$ is denoted by $\text{prefix}(\pi, a_j)$.

**Running Example** Our running example is based on the IPC Rovers domain. A rover must collect up to three samples $S_0, S_1, S_2$ and upload the data to a relay satellite. The rover can perform three different actions: move between locations, take a sample when it is at the corresponding location, and upload the collected data at $l_0$ or $l_3$. The road map and the initial location of the rover are depicted on the left in Figure 1.

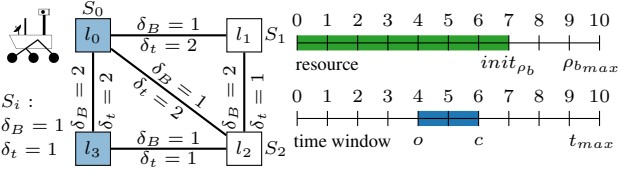

Figure 1: Running Example

## 2.2 Planning with Consumed Resources

A *consumed* resource $\rho$ with domain $\mathcal{D}(\rho) = [0, \rho_{max}] \subset \mathbb{N}$ has an initial value $init_\rho \in \mathcal{D}(\rho)$ and a function $\delta_\rho : A \mapsto \mathbb{N}$ that maps each action to the amount of resource consumed by that action. A state represents a complete assignment to $V \cup \{\rho\}$. Action $a$ is applicable in state $s$ if $\text{pre}_a \subseteq s$ and the remaining value of $\rho$ is sufficient to execute the action $s(\rho) \geq \delta_\rho(a)$. Applying $a$ in $s$ decreases the resource by $\delta_\rho(a)$: $s[[a]](\rho) = s(\rho) - \delta_\rho(a)$. The amount of resource $\rho$ consumed by an action sequence $\pi$ is given by $con(\pi) = \sum_{a \in \pi} \delta_\rho(a)$. An extension to multiple resources is defined accordingly, where the set of all resources is denoted by $R$. In our running example, there is battery $\rho_b$ as a resource.

## 2.3 Planning with Simple Time Windows

We restrict ourselves to a concept of time that can be compiled to classical planning. This means discrete time units and no parallel execution of actions. A (start) time window is a tuple $W = (A_W, o, c)$ with $0 \leq o \leq c \leq t_{max}$. The application of the actions in $A_W \subset A$ is constrained by the opening time $o$ and closing time $c$. The function $\delta_t : A \mapsto \mathbb{N}$ maps each action to its execution duration. The passed time units are represented by the variable $t$ with domain $\mathcal{D}(t) = [0, t_{max}]$. A state represents a complete assignment to $V \cup \{t\}$. Action $a$ is applicable in state $s$ if $\text{pre}_a \subseteq s$ and if $a \in A_W$ then $o \leq s(t) \leq c$. Applying $a$ in $s$ increases the passed time by $\delta_t(a)$: $s[[a]](t) = s(t) + \delta_t(a)$. The execution duration of an action sequence $\pi$ is given by $dur(\pi) = \sum_{a \in \pi} \delta_t(a)$ and the execution time point of action $a$ in $\pi$ by $exec(\pi, a) = dur(prefix(\pi, a))$. An extension to multiple time windows is defined accordingly, where the set of all time windows is denoted with $\mathcal{W}$. In our example, there is one time window $W_U = (\{\text{upload}(S_i) \mid i \in \{0, 1, 2\}\}, 4, 6)$ which allows to upload data to the relay only between the time points 4 and 6.

## 2.4 Explanation Framework

In [Eifler *et al.*, 2020a; Eifler *et al.*, 2020b] $G^{\text{soft}}$ represents a set of plan properties, specifically LTLf plan-preference formulas compiled into soft-goal facts [Baier and McIlraith, 2006; Edelkamp, 2006]. The framework uses **conflicts** between these plan properties to generate answers to the users question. The soft-goals $X, Y \subseteq G^{\text{soft}}$ conflict each other if all plans of $\tau$ that achieve all $g \in X$, do not achieve all $g \in Y$. The strongest dependencies of this kind are given by the **minimal unsolvable goal subsets** (MUGS) $X \cup Y = G \subseteq G^{\text{soft}}$ where $G$ cannot be achieved but every $G' \subsetneq G$ can. The set of all MUGS for a task $\tau$ is denoted by $\text{MUGS}(\tau)$.

# 3 Conflict Explanation Through Relaxations

We provide explanations for a soft-goal conflict based on minimal constraint relaxations under which the conflict disappears. In the following sections, we define the relaxations that are considered and how we identify explanations based on a given set of relaxed tasks.

## 3.1 Relaxation Orders

The most general property of an abstraction or relaxation $T'$ of a planning task $T$ is that all plans of $T$ are preserved [Culberson and Schaeffer, 1998; Edelkamp, 2001; Seipp and Helmert, 2013]. This gives the most general definition of a relaxed task as:

**Definition 1** (Relaxed Task). *Let $T$ be a planning task and $\Pi(T)$ the set of all possible plans for task $T$. Then $T'$ is a relaxed task of $T$ (denoted by $T \sqsubseteq T'$) iff $\Pi(T) \subseteq \Pi(T')$.*

Our explanation approach and algorithms make no further assumptions about the specific implementation of relaxation. To compute the explanation for a conflict $C \in \text{MUGS}(T)$, we assume a final set of relaxed tasks $\mathbb{T}$ for $T$, where $T \in \mathbb{T}$ and for all $T' \in \mathbb{T} : T \sqsubseteq T'$, is given. For $\mathbb{T}$ the relation $T_i \sqsubseteq T_j$ represents a partial order, which we will use to define a *minimal* relaxed task that resolves $C$. The partially ordered set $\hat{\mathcal{T}} = (\mathbb{T}, \sqsubseteq)$ we call in the following a *relaxation order* for $T$. The functions $C_U(T')$ and $C_L(T')$ denote the upper and lower covers of $T'$ within $\hat{\mathcal{T}}$. Given a partially ordered set $S$ then the *upper cover* of an element $e \in S$ is the set $C_U(e) = \{e' \in S \mid e' > e \ \wedge \nexists e'' \in S : e' > e'' > e\}$, and

the *lower cover* the set $C_L(e) = \{e' \in S \mid e' < e \land \nexists e'' \in S : e' < e'' < e\}$. One of our algorithms additionally assumes, that $\hat{\mathcal{T}}$ has a supremum.

## 3.2 Resource and Time Constraint Relaxations

Next we instantiate the above with resource and time window constraint relaxations.

### Resource Constraint Relaxations

A task with consumed resources can be relaxed by increasing the initial resource value.

**Definition 2** (Resource Relaxed Task). *Let $T = (\tau, R)$ be a planning task $\tau$ with resources $R$. Then a resource relaxed task for resource $\rho \in R$ is defined as $T' = (\tau, R')$ where $\rho$ is replaced by resource $\rho'$ with $\mathcal{D}(\rho) = \mathcal{D}(\rho') = [0, \rho_{max}]$, $\delta_{\rho'} = \delta_\rho$ and $\rho_{max} \geq init_{\rho'} \geq init_\rho$.*

A resource relaxed task indeed represents a relaxed task according to Definition 1:

**Proposition 1.** *Let $T'$ be a resource relaxed task of $T$. Then, $\Pi(T) \subseteq \Pi(T')$.*

*Proof sketch*: $\Pi(T) \subseteq \Pi(T')$, because every action sequence $\pi = a_0 \cdots a_n$ applicable in $I$ of $T$ is also applicable in $I'$ of $T'$. For all actions $a_i \in \pi$ with $\pi_i = \textit{prefix}(\pi, a_i)$, $s = I[[\pi_i]]$, $s' = I'[[\pi_i]]$ and $c = con(\pi_i)$, $a_i$ is applicable in $s'$ because $a_i$ is applicable in $s$ and $s(V) = s'(V)$ and $init_\rho - c = s(\rho) < s'(\rho') = init_{\rho'} - c$.

Making the application of an action cheaper by reducing $\delta_\rho$ is another way to relax a task with respect to a resource. This is almost equivalent to increasing the initially available resource, given that the resource is exhaustible and the action appears once in the plan. We use increasing resource availability as a proxy for any reduction in resource consumption.

Using the set $\mathbb{T}_\rho$ of all resource relaxed tasks of task $T$ for $\rho$, we get a well-defined relaxation order $\hat{\mathcal{T}}_\rho = (\mathbb{T}_\rho, \sqsubseteq)$ for $T$. Since all $T_i \in \mathbb{T}_\rho$ are exclusively distinguished by $init_{\rho_i}$ we have $T \sqsubseteq T'$ iff $init_\rho < init_{\rho'}$, which results in a total order for $\mathbb{T}_\rho$. The task $T'$ where $init_{\rho'} = \rho_{max}$ represents the supremum of $\hat{\mathcal{T}}_\rho$. The upper/lower cover of $T' \in \mathbb{T}_\rho$ is the relaxed task where the initial resource value is one unit larger/smaller than in $T'$.

For our running example, we have four relaxed tasks for the battery $\mathbb{T}_{\rho_b} = \{T_i \mid i \in \{7, 8, 9, 10\}\}$, where in $T_i$ the initial battery level is $i$.

### Time Constraint Relaxations

A task with simple time windows can be relaxed by increasing the time window, either by decreasing the opening time or by increasing the closing time.

**Definition 3** (Time-Window Relaxed Task). *Let $T = (\tau, \mathcal{W})$ be a planning task $\tau$ with simple time windows $\mathcal{W}$. Then a relaxed task for time window $W = (A_W, o, c) \in \mathcal{W}$ is defined as $T' = (\tau, \mathcal{W}')$ where $W$ is replaced by $W' = (A_W, o', c')$ with $0 \leq o' \leq o \leq c \leq c' \leq t_{max}$.*

A time-window relaxed task indeed represents a relaxed task according to Definition 1:

**Proposition 2.** *Let $T'$ be a time-window relaxed task of $T$. Then, $\Pi(T) \subseteq \Pi(T')$.*

*Proof sketch*: $\Pi(T) \subseteq \Pi(T')$, because every action sequence $\pi = a_0 \cdots a_n$ applicable in $I$ of $T$ is also applicable in $I'$ of $T'$. For all actions $a_i \in \pi$, $exec(\pi, a_i)$ is the same in both tasks and with $\pi_i = \textit{prefix}(\pi, a_i)$, $s = I[[\pi_i]]$ and $s' = I'[[\pi_i]]$, $a_i$ is applicable in $s'$ because $a_i$ is applicable in $s$ and if $a_i \in A_W$ then $o' \leq o \leq exec(\pi, a_i) \leq c \leq c'$.

An alternative approach to relax a task with respect to time constraints is the reduction of the execution time of an action by decreasing $\delta_t$. However, in addition to affecting multiple time windows, handling the explosion of possible relaxed tasks is not trivial, which is why we leave this for future work.

The subsumption relation of the intervals $[o', c']$ for time window $W$ yields the partial order for $\hat{\mathcal{T}}_W = (\mathbb{T}_W, \sqsubseteq)$, where $\mathbb{T}_W$ is the set of all time-window relaxed tasks of $T$ with respect to $W$. The task with $W' = (A_W, 0, t_{max})$ is the supremum of $\hat{\mathcal{T}}_W$. The upper/lower cover of $T' \in \mathbb{T}_W$ are the relaxed tasks where either $o$ is decreased/increased or $c$ is increased/decreased by one compared to $T'$.

For our running example we have 25 different relaxed tasks for the upload window $\mathbb{T}_{W_U} = \{T_{i,j} \mid i \in \{0, 1, 2, 3, 4\} \land j \in \{6, 7, 8, 9, 10\}\}$, where in $T_{i,j}$ the opening time is at $i$ and the closing time at $j$.

## 3.3 Conflict Explanation

We aim to generate explanations for the conflicts in $MUGS(T)$. Given a relaxation order, we can now define for each conflict whether a task is minimally relaxed for it.

**Definition 4** (Minimally Relaxed Task). *Let $\hat{\mathcal{T}} = (\mathbb{T}, \sqsubseteq)$ be a relaxation order for task $T$ and $C \in MUGS(T)$ a conflict. Then $T' \in \mathbb{T}$ is minimally relaxed for $C \notin MUGS(T')$ if for all $T'' \in \mathbb{T} : T'' \sqsubset T' \to C \in MUGS(T'')$.*

Thus, a minimally relaxed task for conflict $C$ is one in which $C$ is not a conflict, but for all less relaxed tasks it is. All conflicts in $MUGS(T)$, for which $T' \in \hat{\mathcal{T}}$ is minimally relaxed are denoted by mr-$MUGS(\hat{\mathcal{T}}, T')$. The explanation for a conflict in $MUGS(T)$ can then be defined as:

**Definition 5** (Conflict Explanation). *Let $T$ be a task with conflict $C \in MUGS(T)$ and $\hat{\mathcal{T}}$ a relaxation order for $T$. Then the set of all minimally relaxed tasks for $C$, $E(\hat{\mathcal{T}}, C) = \{T' \mid C \in$ mr-$MUGS(\hat{\mathcal{T}}, T')\}$, is the conflict explanation for $C$.*

To illustrate the explanation for conflict $C = \{S_0, S_2\}$ in our running example, we use the diagram in Figure 2. The minimal relaxed tasks and therefore the explanations are given as $E = \{T_{1,6}, T_{4,7}\}$: "Sample $S_0$ and $S_2$ can not both be uploaded, because the upload window, needs either to start 3 time units earlier or to end 1 time unit later".

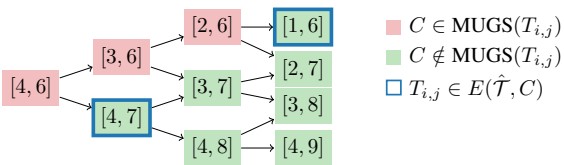

Figure 2: Part of hasse diagram for the time relaxed tasks of the running example. $[i_1, j_1] \to [i_2, j_2]$ means $T_{i_1,j_1} \sqsubseteq T_{i_2,j_2}$. $T_{3,7} \notin E(\hat{\mathcal{T}}, C)$ because $T_{4,7} \sqsubset T_{3,7}$.

**Algorithm 1** Internal Constraint Reuse (ICR)

1: Given: relaxation order $\hat{\mathcal{T}}$, heuristic $h$
2: **function** COMPUTEMSGS($\hat{\mathcal{T}}, h$)
3:     $\mathcal{M} \leftarrow \{\}$                             ▷*map of MSGSs*
4:     **while** HASNEXT($\hat{\mathcal{T}}$) **do**
5:         $\hat{T} \leftarrow$ NEXT($\hat{\mathcal{T}}$)                ▷*current relaxed task*
6:         $\mathcal{M}[\hat{T}] \leftarrow \bigcup_{\hat{T}' \in C_L(\hat{T})} \mathcal{M}[\hat{T}']$     ▷*propagate MSGS*
7:         $\mathcal{O} \leftarrow \{$INIT($\hat{T}$)$\}$        ▷*initial state of relaxed task*
8:         **while** $|\mathcal{O}| \neq 0$ **do**
9:             $s \leftarrow$ NEXT($\mathcal{O}, h$)    ▷*next state according to expansion order*
10:            **if** $G^{\text{hard}} \subseteq s$ **then**            ▷*update MSGS*
11:                $\mathcal{M}[\hat{T}] \leftarrow$ EXTEND($\mathcal{M}[\hat{T}], s \cap G^{\text{soft}}$)
12:            **if** ($G^{\text{hard}} \cup G^{\text{soft}}$) $\subseteq s$ **then**   ▷*check and propagate solvability*
13:                $\forall T' \in \hat{\mathcal{T}} \wedge \hat{T} \sqsubseteq T' : \mathcal{M}[T'] \leftarrow G^{\text{hard}} \cup G^{\text{soft}}$
14:                **break**
15:            $\mathcal{O} \leftarrow \mathcal{O} \cup \{s' \in$ SUCC($\hat{T}, s$) $\mid \neg$PRUNE($\hat{T}, \mathcal{M}[\hat{T}], h, s'$)$\}$
16:     **return** $\mathcal{M}$

---

**Algorithm 2** Search Space Reuse (SSR)

1: Given: relaxation order $\hat{\mathcal{T}}$, heuristic $h$
2: **function** COMPUTEMSGS($\hat{\mathcal{T}}, h$)
3:     $T_s \leftarrow$ SUPREMUM($\hat{\mathcal{T}}$)         ▷*maximal relaxed task*
4:     $\mathcal{M} \leftarrow \{\}$                          ▷*map of MSGSs*
5:     $\mathcal{F} \leftarrow \{\}$                        ▷*map of search frontiers*
6:     $\mathcal{O} \leftarrow$ INIT($T_s$)        ▷*initial state of maximally relaxed task*
7:     $\hat{T} \leftarrow$ NEXT($\hat{\mathcal{T}}$)              ▷*current relaxed task*
8:     **while** True **do**
9:         **while** $|\mathcal{O}| \neq 0$ **do**
10:            $s \leftarrow$ NEXT($\mathcal{O}, h$)    ▷*next state according to expansion order*
11:            **if** $G^{\text{hard}} \subseteq s$ **then**          ▷*update MSGS*
12:                $\mathcal{M}[\hat{T}] \leftarrow$ EXTEND($\mathcal{M}[\hat{T}], s \cap G^{\text{soft}}$)
13:            **if** ($G^{\text{hard}} \cup G^{\text{soft}}$) $\subseteq s$ **then**   ▷*check and propagate solvability*
14:                $\forall T' \in \hat{\mathcal{T}} \wedge \hat{T} \sqsubseteq T' : \mathcal{M}[T'] \leftarrow G^{\text{hard}} \cup G^{\text{soft}}$
15:                **break**
16:            $S_{suc} \leftarrow \{s' \in$ SUCC($T_s, s$) $\mid \neg$PRUNE($T_s, \mathcal{M}[\hat{T}], h, s'$)$\}$
17:            $\mathcal{F}[\hat{T}] \leftarrow \mathcal{F}[\hat{T}] \cup \{s', \in S_{suc} \mid \neg$APPL($\hat{T}, \pi(s')$)$\}$
18:            $\mathcal{O} \leftarrow \mathcal{O} \cup \{s' \in S_{suc} \mid$ APPL($\hat{T}, \pi(s')$)$\}$
19:     **if** $\neg$HASNEXT($\hat{\mathcal{T}}$) **then**
20:         **return** $\mathcal{M}$
21:     $\hat{T} \leftarrow$ NEXT($\hat{\mathcal{T}}$)              ▷*current relaxed task*
22:     $\mathcal{M}[\hat{T}] \leftarrow \bigcup_{\hat{T}' \in C_L(\hat{T})} \mathcal{M}[\hat{T}']$     ▷*propagate MSGS*
23:     $F_C \leftarrow \bigcup_{\hat{T}' \in C_L(\hat{T})} \{s' \in \mathcal{F}[\hat{T}'] \mid \neg$PRUNE($T_s, \mathcal{M}[\hat{T}], h, s'$)$\}$
24:     $\mathcal{F}[\hat{T}] \leftarrow \{s' \in F_C \mid \neg$APPL($\hat{T}, \pi(s')$)$\}$
25:     $\mathcal{O} \leftarrow \{s' \in F_C \mid$ APPL($\hat{T}, \pi(s')$)$\}$

# 4 Internal Constraint Reuse (ICR)

In the following, we introduce two algorithms that given a relaxation order $\hat{\mathcal{T}}$ compute the **maximal solvable goal subsets** (MSGS) for each task. The MSGS can then be used to compute the mr-MUGS of each task as follows.

**From MSGS to mr-MUGS** A MSGS is a soft-goal subset $G \subseteq G^{\text{soft}}$ where $G$ can be achieved but every $G' \supsetneq G$ cannot. Given MSGS($T'$) for $T' \in \mathbb{T}$ we compute mr-MUGS($\hat{\mathcal{T}}, T'$) in two steps. First, we compute MUGS($T'$) by performing a bottom-up tree search over all subsets of $G^{\text{soft}}$ and use the MSGS as a fast solvability check as introduced by [Eifler *et al.*, 2020b]. Then, the MUGS for which $T'$ is minimally relaxed are computed as mr-MUGS($\hat{\mathcal{T}}, T'$) = MUGS($T$) $\cap$ $((\bigcap_{T'' \in C_L(T')} \text{MUGS}(T'')) \setminus \text{MUGS}(T'))$.

**MSGS Computation** Eifler et al. [2020b] compute the MSGS for a task by exhaustively exploring the state space while tracking all reached MSGS. To reduce the search space size they introduce a pruning function, which prunes all states from which no superset of the current MSGS is reachable.

Extending this algorithm, given a relaxation order $\hat{\mathcal{T}} = (\mathbb{T}, \sqsubseteq)$ for $T$, we compute the MSGS for all $T' \in \mathbb{T}$ by iterating over $\mathbb{T}$ according to the partial ordering, starting with $T$, and computing the MSGS for each task individually. Since all plans are preserved, for all $T_i, T_j \in \mathbb{T}$ with $T_i \sqsubseteq T_j$, all soft-goals $G \subseteq G^{\text{soft}}$ that are reachable in $T_i$ are also reachable in $T_j$. Thus, the MSGS of $T_i$ can be propagated to $T_j$.

**Pseudo Code of ICR** The pseudo code of the *Internal Constraint Reuse* (ICR) algorithm is given in Algorithm 1. The underlying search algorithm for the state space exploration of one task is depth first search (DFS) guided by a heuristic (see [Eifler *et al.*, 2020b]). This is abstracted by the function NEXT($\mathcal{O}, h$), which mimics the expansion order of DFS. $\mathcal{M}$ (line 3) is a map from task to a set of soft-goal subsets storing the MSGS for each task. The aforementioned propagation of MSGS is realized by initializing $\mathcal{M}[\hat{T}]$ with all MSGS reached in the lower cover of $\hat{T}$ (line 6). Iterating over the relaxed tasks according to the partial ordering is represented by the functions HASNEXT($\hat{\mathcal{T}}$) and NEXT($\hat{\mathcal{T}}$). The order of incomparable elements is resolved randomly.

$\mathcal{M}[\hat{\mathcal{T}}]$ is only updated (line 11) in case no superset has already been reached: EXTEND($M, G$) returns $M$ if there is a $G' \in M : G \subseteq G'$ and $\{G' \in M | G' \not\subseteq G\} \cup \{G\}$ otherwise. The generation of successor states of state $s$ according to the semantics of task $T$ (SUCC($T, s$)) is based on standard progression (see Sections 2.2 and 2.3). States are pruned (line 15) if no superset of soft-goals or the hard goal cannot be reached: PRUNE($T', M, h, s$) returns true if $G^{\text{hard}} \not\subseteq R \vee \exists G \in M : R \cap G^{\text{soft}} \subset G$, where $R$ is the set of facts reachable from state $s$ in task $T'$ according to heuristic $h$ and false otherwise. This can for example be realized by checking for each fact whether its $h^{\max}$ [Haslum and Geffner, 2000] estimation is finite, for more details we refer to [Eifler *et al.*, 2020b]. If a state satisfies all hard and soft goals the search for the current task can be terminated early (line 12). All more relaxed tasks are also solvable, so their MSGS are updated accordingly (line 13). Tasks whose MSGS have already been determined are skipped by HASNEXT/NEXT.

# 5 Search Space Reuse (SSR)

ICR causes overhead, because equivalent states are generated multiple times in the separate search spaces. In addition to the MSGS, it can be beneficial to reuse the search space too.

Since all plans are preserved in the relaxations $\mathbb{T}$ of task $T$, for all $T' \in \mathbb{T}$ and the most relaxed task $T_s = supremum(\hat{\mathcal{T}})$ holds $(\bigcup_{T'' \in C_L(T')} S_{T''}) \subseteq S_{T'}$, where $S_{T'}$ are the states reachable from the initial state $I_s$ of $T_s$ by plans of $T'$. Thus, we can base the computation of the MSGS for all relaxed tasks on the search space of $T_s$. We begin by exploring the reachable state space for $T$. All states that are generated, but which are not reachable in $T$, are stored in a search frontier. To decide whether a state $s$ is reachable in a task $T'$, we check whether the action sequence $\pi(s)$ leading to $s$ is applicable in $T'$ (APPL($T', \pi(s)$)). In our example, states reached by

paths consuming more than 7 energy units are not reachable in $T_7$ and are therefore stored in the frontier. In subsequent iterations, the search frontiers of less relaxed tasks are further extended for more relaxed tasks. For example, for $T_8$ the frontier states of $T_7$ are further extended. This limits the states generated for each task to the newly reachable states.

**Pseudo Code of SSR** The pseudo-code of the *Search Space Reuse* (SSR) algorithm is depicted by Algorithm 2. Unless explicitly mentioned, the algorithm parts work as described for Algorithm 1. The map $\mathcal{F}$ (line 5) stores for each task $\hat{T}$ the states which were generated during the search for $\hat{T}$ but were not reachable (line 17). In the first iteration, the openlist is initialized with the initial state of the most relaxed task $T_s$ (line 6). In each subsequent iteration, it is initialized with the states in $\mathcal{F}$ of all tasks in the lower cover of $\hat{T}$ that are reachable in $\hat{T}$ (line 23-25). States are pruned by following the same approach as in Algorithm 1 (line 16/23). However, instead of basing the pruning on the current relaxed task $\hat{T}$ it is based on the most relaxed task $T_s$, since otherwise states that might be reachable in more relaxed tasked, would be pruned.

## 6 Theoretical Comparison

The propagation of MSGS can improve the pruning function, which is beneficial to both ICR and SSR. Reusing the search space in SSR reduces duplicate work, but states are only pruned based on the reachability in the most relaxed task, not the current task. We compare the overall number of generated states by each algorithm as a measure to decide whether they are exponentially separated. As the baseline algorithm, we consider ICR without the propagation of MSGS.

**Definition 6** (Exponential Separation). *Let $\{T^n | n \in \mathbb{N}\}$ be a family of planning tasks of size (number of facts and actions) polynomially related to $n$ and $S(X)$ the number of states generated by search method X. Then, search method X is exponentially separated from search method Y iff $|S(Y) - S(X)|$ is exponential in $n$.*

To give a family of planning tasks to prove the exponential separations of the algorithms we consider a planning task, where a robot has to visit different locations. The robot's movement is restricted by the resource $\rho$, which can have the values $\{0, 1, 2\}$, with initial value 1. Moving between connected locations consumes the amount of resources depicted in the maps in Figure 3. There is one location annotated with $K$ which holds a set of $n$ keys. The robot can pick up one key at a time (without using any resources) if it is in the same location as the key. To take the dashed connection the robot has to hold all keys. Since the robot can pick up any combination of keys, there can be exponentially many search states.

In the following examples, the pruning function uses the $h^2$ heuristic [Haslum and Geffner, 2000] to decide reachability.

**Theorem 1.** *ICR and SSR are exponentially separated from the baseline.*

**Example** Consider the map depicted on the left in Figure 3. In the first iteration with $init_\rho = 1$ ICR and the baseline generate 2 states (R at $L_0$ and $L_1$). SSR, with $init_\rho = 2$, generates the same two states and 2 additional states (R at $L_2$ and $L_3$),

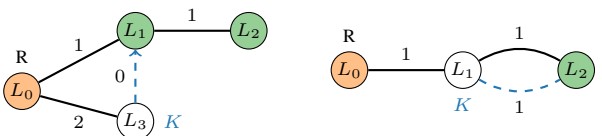

Figure 3: left: Map of separation of ICR and SSR from baseline, initial location: $L_0$, goal: visit $L_1$ and $L_2$. right: Map of separation of ICR from SSR, initial location $L_0$, goal: visit $L_2$.

which are not reachable and stored in the frontier. For both ICR and SSR MSGS $= \{\{L_1\}\}$ is propagated. In the next iteration of ICR ($init_\rho = 2$), moving to $L_3$ is pruned because no new locations are reachable from there. The same holds for SSR. This leads to $2 + 3$ and $4 + 1$ states for ICR and SSR respectively. For the baseline, the reachability of $L_1$ is not propagated and moving to $L_3$ is not pruned. Thus, we get $2 + 3 + 2 * 2^n$ states, for picking up any combination of keys.

**Theorem 2.** *ICR is exponentially separated from SSR.*

**Example** Consider the map depicted on the right in Figure 3. In the first iteration with $init_\rho = 1$, ICR generates only one state. Moving to $L_1$ is pruned because $h^2$ recognizes that $L_2$ is not reachable with $\rho = 1$. In SSR, the weaker constraint $init_\rho = 2$ prevents pruning $L_1$ and picking up any combination of keys. Thus, $1 + 2^n$ reachable and $2^n$ (at $L_2$ with any combination of keys) unreachable states are generated. In the last iteration in ICR with $init_\rho = 2$ visiting $L_2$ via the upper connection and extending the MSGS to $\{\{L_2\}\}$ leads to early termination. The same holds for SSR. This results in $1 + 3$ states for ICR and $1 + 2 * 2^n$ for SSR.

## 7 Experiments

We implemented both algorithms in the Fast Downward planning system [Helmert, 2006], extending the code base of Eifler et al. [2020b] and using $h^{\max}$ as a base heuristic for the pruning function[1]. The experiments were run on Intel E5-2660 machines running at 2.20 GHz, with a time (memory) limit of 2h (4GB) per benchmark instance.

**Benchmark** Our benchmark consists of 4 resource-constraint domains (Blocksworld, NoMystery, Rovers R, TPP) and 3 domains (Parent's Afternoon, Rovers T, Satellite) with time constraints. The former part builds on the resource constraint benchmark by Eifler et al. [2020b]. In each instance there are two individual resources $R$. For each resource $\rho \in R$ we generated one benchmark instance, scaling $init_\rho$ between 0 and two times the initial value in the original instance. Rover T and Satellite are extension of the IPC domains with data upload windows for Rovers and time windows to take the images for Satellite. Parents' Afternoon [Eifler *et al.*, 2022] models a parent's afternoon routine, including shopping and family member activities. The execution of these activities is constraint by time windows. For each time window $W \in \mathcal{W}$ we generated one benchmark instance. For Parent's Afternoon and Satellite, each time window is relaxed between its original size and the maximal value of the time

---

[1]The source code and the benchmark are available at: https://github.com/XPP-explainable-planning

variable domain. For Rovers the relaxation of an upload window is additionally bounded by the other upload windows. Each benchmark instance has up to 5 plan properties that, for example, restrict the order in which two goal facts are to be achieved. All plan properties and the original goal facts of the instance are soft goals. There are no hard goals.

## 7.1 Evaluation

The coverage results are shown in Table 1. An instance is considered to be solved, when the MUGS for all relaxed tasks are computed. Comparing the ICR to the baseline shows propagating the MSGS increases the coverage in 5 domains, while not decreasing it in any. SSR solves more instances in 4 domains, while it is worse than the baseline in 2. ICR clearly has the advantage over SSR in the resource domains, while it is the opposite in the time constraint domains.

| | domain | # | base | ICR | SSR |
|---|---|---|---|---|---|
| resource | Blocksworld | 40 | 18 | 18 | **19** |
| | NoMystery | 50 | 12 | **24** | 9 |
| | Rovers R | 40 | **20** | **20** | 15 |
| | TPP | 30 | 11 | **19** | 9 |
| time | Parent's A. | 72 | 35 | 37 | **53** |
| | Rovers T | 138 | 47 | 53 | **96** |
| | Satellite | 198 | 123 | 130 | **144** |

Table 1: Coverage (number of instances solved); ICR and SSR: algorithms introduced in Section 4 and 5; base: ICR without propagation of MSGS. Best result for each domain is highlighted in bold.

The increase in reachable states caused by relaxing a time window is usually much smaller than for a resource. Increasing a time window only adds few more times at which a single action $a \in A_W$ could start. However, as $a$ is also constrained by all other time dependent actions, there may not be many added reachable states. In contrast, relaxing a resource allows you to add new actions and increases the number of action orderings. This is in favor for SSR, because it only considers the newly reachable states. A comparison of the number of expansions each algorithm requires per relaxed task, as depicted in Figure 4, confirms this assumption. In the time constraint domains SSR expands more states than ICR in the first task, but has many fewer expansions than ICR thereafter. In the resource-constraint task, the stronger pruning function in ICR is advantageous for a wider span of relaxed tasks, such that SSR only needs fewer expansions in more relaxed tasks.

Problems may not be solved either due to the exhaustion of the time or the memory limit. For Blocksworld and all time constraint domains, all algorithms ran out of time. For the other resource constraint domains SSR failed due to the memory limit. In TPP ICR failed due to the time limit and in rovers due to the memory. In Nomystery failure of ICR- was cased by about 25% timeouts and 75% memory limit exhaustion. Overall, timeout is most common. This could be addressed by parallelization of tasks without a strict order.

## 8 Related Work

Sreedharan et al. [2019] explain the unsolvability of a task by identifying necessary subgoals of relaxed tasks, that are unachievable in the original task. However, due to use of relaxations based on projections on subsets of variables, this

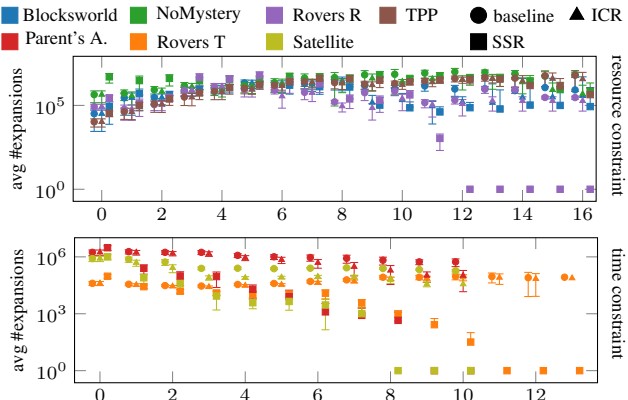

Figure 4: Comparison of average number of expansions over commonly solved task. Error bars represent the 95% confidence intervals. top: resource constraint domains, x value corresponds $init_\rho$; bottom: time constraint domains, x value corresponds to size difference of the relaxed time window to the original one.

approach is not suitable for quantifying the relaxation necessary to make the task solvable. An 'excuse' for unsolvability, as defined by [Göbelbecker et al., 2010], is a series of value changes in the initial state and additional objects to make a task solvable. Their approach does not provide an explanation for a specific conflict, but explains why the task is not solvable and focuses on pointing out errors in the model description. The scheduling system by Agraval et al. [2020] provide information about constraint relaxations for not scheduled activities. Their main focus is to identify all unmet constraints of an activity and present them alongside the schedule to facilitate user review. Their analysis does not yet include any reasoning on the extent to which a constraint needs to be relaxed to schedule the activity. The unsolvability certificates provided by the proof system of Eriksson et al. [2017; 2018] are not intended to be human-readable and do not provide information on how the task could be made solvable.

The resource and time window constraints we consider here, can be compiled to classical planning and relaxation can be represented by domain abstractions [Domshlak et al., 2009]. Resources constraint relaxation could additionally be simulated by cost bound relaxation in classical planning with multiple cost functions [Katz et al., 2019; Altman, 1999].

## 9 Conclusion

Our approach addresses the question why soft-goal conflicts exist by identifying the minimal relaxation under which a conflict disappears. Combined with the work of Eifler et al.[2020a; 2020b], this provides an explanation framework that can explain trade-offs between soft goals by identifying not only conflicts, but also options for resolving them. This not only helps to better understand why a conflict exists, but also whether it can be resolved. In addition it enables the user to evaluate the trade-offs and benefits of a relaxation.

Future work includes the evaluation in an application setting and the automatic identification of relevant relaxations for a user and conflict.

## Acknowledgments

This material is based upon work supported by the Air Force Office of Scientific Research under award number FA9550-18-1-0245, and by the German Research Foundation (DFG) under grant 389792660 as part of TRR 248 (see https://perspicuous-computing.science).

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
