# OpenReview forum: "Explaining Soft-Goal Conflicts through Constraint Relaxations"
_icaps-conference.org/ICAPS/2022/Workshop/XAIP — XAIP 2022_

### Official Review · Reviewer_1nMH · 2022-04-28
**Interesting paper**

**Rating:** 7
**Confidence:** 4

**Review:**

- **Paper Summary** - The paper extends the idea of providing plan space explanations in terms of conflicts to providing extra insight into why these conflicts cannot be achieved. Previous work suggests to provide an explanation for the question “Why a conjunction A of goals is not achieved by $\pi$?” by providing conflicts (the minimal unsolvable soft-goal subsets) if A were to be enforced. The authors in this paper say that such explanations do not explain the conflicts themselves as in why a set of soft-goals not be jointly achieved. Using the underlying constraints (resource and time constraints) on plans in the task at hand, they provide explanations in the form of minimally relaxed tasks under which the conflict disappears. The authors propose a baseline and two algorithms that leverage the information at hand to improve over the baseline. They provide a theoretical comparison where they show that the two algorithms outperform the baseline exponentially with respect to the number of generated states. Finally they provide a computational evaluation of these 3 algorithms on several IPC domains with time and resource constraints.
- **Relevance to XAIP** - This work is definitely relevant to the workshop as this work is connected to inference reconciliation and actionable explanations.
- **Strengths**
    - The authors propose two neat algorithms that act as extensions to the previous work to provide better insights to a user’s explanatory question.
    - The authors provide a valid theoretical comparison between the algorithms and the proposed baseline.
    - The authors present computational results which reflect the theoretical comparison.
    - The authors formalisms are sound (as far as I could understand)
- **Suggestions and Questions**
    - Some aspects in the paper could be clearer, specifically,
        - In the proof sketches for proposition 1 and proposition 2, why a sequence of actions in T can be applicable in T’ needs to be clearer
        - Even though the alternative approach for relaxing a resource constrained task made sense to me, the alternative approach for a time constrained task and how it might lead to a explosion of possible relaxed tasks could be elaborated a bit more for the reader to better understand the approach.
    - *(Question)* In Figure 3, shouldn’t the goal visit for the right hand side of the figure be $L_2$ instead of $L_3$ ? If it is just $L_3$, the how would $L_2$ be part of the MSGS?
    - For explaining illustration in Figure 2, it would probably be better to reiterate relevant parts of the running example to better help the reader.
    - To better understand the statement made about a higher increase in the reachable states for relaxed resource constrained problems, in the resource constraint graph in Figure 4, it might be helpful to have the y-axis zoomed in and declutter the graph a bit.
    - Figure 1 can be larger in size for better readability of the resource and time constraints written on the graph edges.
    - It might be interesting to evaluate domains consisting of both time and resource constraints.
    - *(Question)* Did you encounter a case where there was no relaxation of a task in which the conflicts disappeared?
- **Final Remarks** - To conclude, I think that this paper is relevant to the workshop and should be accepted.

---

### Official Review · Reviewer_Xetr · 2022-04-30
**Interesting manuscript on conflict explanation**

**Rating:** 7
**Confidence:** 4

**Review:**

The manuscript contributes to generating explanations for soft-goal conflicts in planning. The topic is clearly relevant to XAIP. The authors propose to explain soft-goal conflicts by relaxing resource and time constraints until the to-be-explained soft-goal conflict does not occur anymore and report the relaxation as a counterfactual explanation (viz., if there had been n more resources available, the conflict would not have occurred). The relaxations are well described, and two algorithms for generating the explanations are proposed and experimentally evaluated.

Some remarks:
- I wonder whether the relaxation of resources and time could be unified. Couldn't time also be modeled as a resource?
- Not all soft-goal conflicts are due to a shortage of resources or time. What explanation should a user receive in those cases?
- I'd slightly disagree with the comment on the related work by Göbelbecker et al. Their approach also generates a counterfactual explanation for why a conflict occurs (unsolvability) w.r.t. conditions in the initial state, which are useful to understand what to do in order to make the problem solvable (e.g., open a closed door so that the robot can reach its goal).
- At various places in the text, there is an unresolved reference Eif20.

Pro:
- Relevant topic to XAIP
- Well written
- Experimental evaluation of the algorithms

Con:
- Some points could be added to the discussion (see above)

---

### Meta-Review · Program_Chairs · 2022-04-30

**Recommendation:** Accept
**Confidence:** 5

**Metareview:**

Both reviewers note that the paper presents a novel and sound approach to addressing the problem of explaining conflicts between soft goals. The reviewers note some clarity issues, which hopefully the authors would address in the future version. In terms of the next steps an interesting suggestion was to look at a way to unify the different relaxation. Also echoing, one of the reviewers comments the authors might also want to investigate how the proposed method is related to the notions of counterfactual explanations and actionable explanations as studied in the wider XAI literature.

In summary, I am quite confident that the paper would be a good fit for XAIP and would recommend accepting it.

---

### Decision · Program_Chairs · 2022-04-30

Accept